# Emergence of Resistance to Integrase Strand Transfer Inhibitors during Dolutegravir Containing Triple-Therapy in a Treatment-Experienced Patient with Pre-Existing M184V/I Mutation

**DOI:** 10.3390/v12111330

**Published:** 2020-11-19

**Authors:** Dominique L. Braun, Thomas Scheier, Ulrich Ledermann, Markus Flepp, Karin J. Metzner, Jürg Böni, Huldrych F. Günthard

**Affiliations:** 1Division of Infectious Diseases and Hospital Epidemiology, University Hospital Zurich, University of Zurich, 8091 Zurich, Switzerland; thomas.scheier@usz.ch (T.S.); karin.metzner@usz.ch (K.J.M.); huldrych.guenthard@usz.ch (H.F.G.); 2Institute of Medical Virology, University of Zurich, 8057 Zurich, Switzerland; boeni.juerg@virology.uzh.ch; 3Private Practice, 8008 Zurich, Switzerland; ulrich.ledermann@hin.ch; 4Center for Infectious Diseases, 8038 Zurich, Switzerland; markus.flepp@hin.ch

**Keywords:** HIV-1, antiretroviral therapy, dolutegravir, virological failure, drug resistance, M184V/I mutation, integrase strand transfer inhibitor

## Abstract

With the current widespread use of dolutegravir in low-income countries, the understanding of the impact of nucleoside reverse transcriptase inhibitor (NRTI-) associated mutations on the efficacy of dolutegravir-containing antiretroviral therapy (ART) is of utmost importance. We describe a rare case of a patient with pre-existing M184V/I mutation and virological failure on a dolutegravir/lamivudine/abacavir regimen with the emergence of integrase strand transfer inhibitor resistance mutations. Additional risk factors, which may have triggered the virological failure, included suboptimal adherence and low nadir CD4+ cell count. This case illustrates that dolutegravir-containing triple-therapy should be prescribed with caution to patients with pre-existing M184V/I mutation and poor efficacy of the reverse transcriptase inhibitor backbone. In addition, this case highlights the need for viral load monitoring in patients on dolutegravir-containing regimens in settings with a high prevalence of the M184V/I mutation such as in low-income countries.

## 1. Introduction

Infection with the human immunodeficiency virus type 1 (HIV-1) can be very efficiently treated with combination antiretroviral therapy (cART), however, this success can be jeopardized by the emergence of HIV-1 drug resistance. Dolutegravir (DTG) is a second-generation integrase strand transfer inhibitor (INSTI) often used as a component of preferred cART due to its favorable properties, including a high genetic barrier to resistance, good tolerability and safety profile, and low potential for drug-drug interactions [1]. In clinical trials at 96 and 148 weeks, no emergence of resistance to INSTI has been reported among patients failing on first-line DTG-containing triple-therapy. However, real-life data from treatment-naïve and treatment-experienced patients on DTG-containing triple-therapy describe a few cases of virologic failure with the emergence of HIV-1 resistance to INSTIs [2,3,4,5,6].

With the current widespread use of DTG in low-income countries, the understanding of the impact of nucleoside reverse transcriptase inhibitor (NRTI-) associated mutations on the efficacy of dolutegravir-containing ART is of utmost importance [7]. The M184V/I mutation is one of the most common NRTI-associated mutations in HIV-1 infected patients and typically selected in patients failing on lamivudine (3TC) or emtricitabine (FTC) containing regimen. The presence of M184V/I reduces the susceptibility to these drugs by more than 100-fold and additionally causes an impaired efficacy to abacavir (ABC). In contrast, it enhances the susceptibility to tenofovir disoproxil fumarate (TDF) and tenofovir alafenamide fumarate (TAF). Recently, a prospective study using data from five large HIV cohorts in four European countries assessed the efficacy of the ABC/3TC/DTG regimen in virologically suppressed, treatment-experienced patients and found no evidence for an impact of previously acquired M184V/I mutation on the incidence of virological failure [8]. Furthermore, no new mutations were observed in reverse transcriptase or integrase after virological failure in any of the patients in whom genotyping was successfully performed. However, due to the short observation time and the low number of virological failure events, the study concluded that additional analyses are required to demonstrate whether these findings remain robust during an extended observation period.

Here we describe a rare case of a patient with a pre-existing M184V/I mutation who failed on the ABC/3TC/DTG regimen with the emergence of resistance to all currently licensed INSTIs including the second-generation INSTIs DTG and bictegravir (BIC).

## 2. Case Description and Results

A 45-year-old man who has sex with men was diagnosed with HIV-1 infection in March 1997. The CD4+ cell count at the time of diagnosis revealed 325 cells per microliter (µL) blood with an HIV-1 viral load of 94,267 copies (cp) per milliliter (mL) plasma. His medical history was remarkable for chronic depression and intermittent illicit drug use. ART consisting of nelfinavir (NFV), 3TC, and stavudine (d4T) was initiated and virus suppression below the limit of quantification of 50 cp/mL plasma was achieved six months later (Figure 1). According to the autovaccination hypothesis that re-exposure to HIV-1 during treatment interruptions may stimulate the HIV-1 specific immune response and lead to low viremia after the withdrawal of ART, the patients were enrolled into the Swiss-Spanish Intermittent Treatment Trial (SSITT) in October 1999 [9]. In brief, in this trial ART was interrupted for 2 weeks, restarted, and continued for 8 weeks. After four such cycles, treatment was indefinitely suspended. During the indefinite treatment interruption period, the viral load increased to a maximum of 25,558 cp/mL and decreased to a minimum of zero copies/mL as measured with a single copy assay when treatment with NFV, 3TC, and d4T was re-initiated. Longitudinal blood samples were analyzed using an in-house allele-specific-PCR (AS-PCR) to determine the frequency of M184V and L90M drug-resistant variants [10]. The M184V mutation was not detectable in the second structured treatment interruption cycle. In the fifth cycle, six time-points were measured, and one of them was positive for the M184V mutation at a low frequency of 0.9% (Figure 1). The amplification of samples at the other short structured treatment interruption cycles and additional time points during the fifth cycle was unsuccessful.

Following the SSITT trial, the patient remained off ART until October 2008. At that time, the CD4+ cell count had decreased to a nadir of 167 cells/µL blood with a viral load of 114,000 cp/mL plasma and the patient had developed candida esophagitis. Prior to developing symptoms, he did not want to reinitiate treatment although this was recommended according to treatment guidelines at that time [11]. In October 2007, a genotypic HIV resistance test using Sanger technology was performed from a blood repository sample taken before the patient started his first ART in 1997. This resistance test revealed infection with HIV-1 subtype B and the presence of the accessory mutations M36I, I62V, and V77I not causing any clinically relevant resistance to antiretroviral drugs. The M184V mutation, which was identified during the SSITT trial, was not present in 1997, suggesting a selection of the M184V mutation during the structured treatment interruption cycles (Figure 1). In October 2008, ART with ritonavir (RTV) boosted lopinavir (LPV) and tenofovir disoproxil fumarate (TDF)/FTC was started. However, the treatment was switched to ritonavir-boosted fosamprenavir (FAPV) with the continuation of TDF/FTC some days later because of severe LPV/RTV-associated diarrhea. With this ART the viral load remained <50 copies/mL plasma. In May 2011, the patient experienced virological failure with an increased viral load of 3390 copies/mL plasma, most likely caused by suboptimal adherence as self-reported by the patient. The genotypic resistance test using Sanger technology revealed the M184I mutation causing high- and low-level resistance to 3TC/FTC and ABC, respectively. In addition, the accessory protease mutations K20R, L10I, I13V, and G16E were detected. ART was switched to RTV-boosted darunavir (DRV) and raltegravir (RGV) with the continuation of TDF/FTC. With this salvage therapy, the viral load was fully suppressed one month later and ART was continued with DRV/TDF/FTC. Within the following 4 years, the viral load remained always below the limit of quantification of <20 cp/mL plasma, but viral RNA was detectable at most time points. In June 2015, the patient demanded to switch to a single-tablet regimen (STR), because of the pill burden of his ART. Hence, ART was switched to the STR DTG/3TC/ABC. Afterward, intermittent low-level viremia up to 383 cp/mL occurred, most likely associated with periods of poor ART adherence. Indeed, the patient reported to miss his ART regularly for several consecutive days, often related to concomitant illicit drug use. During periods of good adherence, however, the viral load decreased to levels <20 cp/mL plasma. In October 2019, when the viral load increased from 45 cp/mL to 183 cp/mL plasma, a genotypic resistance test was performed. This resistance test newly revealed the E138K, Q148R, and R263K INSTI mutations at frequencies of 100% by means of next-generation sequencing (NGS), causing high-level resistance to RAL, EVG, DTG, and BIC. In addition, the prior M184V mutation—but no other NRTI mutations—was detected at a frequency of 100%. The patient’s ART was switched to a salvage regimen consisting of twice-daily RTV-boosted DRV, twice-daily etravirine (ETR), and tenofovir alafenamide fumarate (TAF)/FTC (Figure 1). With this salvage regimen, the viral load returned to levels <20 cp/mL plasma and remained suppressed until present in January 2020.

## 3. Discussion

This is a rare case of a patient with pre-existing M184V/I mutation and virological failure on a DTG/3TC/ABC regimen with the emergence of INSTI resistance mutations. Most likely, the virological failure was triggered by suboptimal adherence and a low nadir CD4+ cell count.

Virological failure with the emergence of resistance to INSTIs on a DTG-based triple-therapy is a rare event with only a few cases published so far [2,3,4,5,6]. Our patient exhibited several features that might have increased his risk for a virological failure: Firstly, the patient re-started his ART following a structured treatment interruption with a low CD4+ cell count of 167 cells/µl blood. Indeed, a CD4+ cell count below 200 cells/µL blood is associated with less favorable virological, immunological, and clinical outcomes in many studies [12]. Secondly, the patient reported suboptimal adherence during several years with detectable low-level viremia and viral blips. Thirdly, the patient harbored the M184V/I mutation, which causes reduced susceptibility to two (3TC, ABC) out of the three components of the DTG/3TC/ABC regimen. A recent European observational study assessed the impact of the M184V/I mutation on the virological failure rate on the DTG/3TC/ABC regimen [8]. Of the 1626 patients included, 137 (8.4%) harbored the genotypically documented M184V/I mutation. Patients with the M184V/I mutation had a lower CD4 nadir and a long history of antiviral treatment. Overall, the study observed a very low virological failure rate (1.3%) after the switch to ABC/3TC/DTG, with no statistically significant difference in the virological failure incidence among patients with or without M184V/I. However, the generalization of the results is limited due to the small sample size and the short observational period (median follow-up 288 days). Furthermore, an indication bias may have occurred as physicians tended to prescribe a single-pill regimen in patients with documented resistance mutations only if they were confident about patients’ adherence [8]. It is convincing that in our case the combination of poor adherence with low-level replication and pre-existing M184V/I mutation, first observed as a low-abundant variant in the SSITT study, led to virological failure and consecutively emergence of INSTI resistance.

The learning points of this case are manifold: Firstly, this case illustrates that patients with a low CD4+ cell count nadir, pre-existing NRTI-associated mutations, and poor adherence are not candidates for a DTG-containing regimen with limited efficacy of the NRTI-backbone, at least not without close monitoring of the viral load. Other INSTIs with a lower resistance barrier than DTG, such as elvitegravir (EVG) in the co-formulation EVG/cobicistat/FTC/TAF, showed to be effective in maintaining viral suppression despite archived M184V/I mutations at week 24 in a prospective open-label study. However, in this study-population the median nadir CD4+ cell count was high (724 cells/µL blood), the sample size was small, and the observational period was rather short [13]. Furthermore, TAF exhibits still full efficacy in the presence of the M184V/I mutation in contrast to ABC. Secondly, our case and others indicate that DTG might have a lower resistance barrier than previously postulated, in particular when given with a not fully active backbone. This finding may have important consequences in light of the current widespread use of DTG in low-income countries. In this setting, pre-existing NRTI-associated mutations—including the M184V/I mutation—are highly prevalent and there is no possibility of close viral load monitoring. In addition, the cut-off of a viral load >1000 cp/mL plasma recommended by the World Health Organization (WHO) for the definition of virological failure will lead to delayed action when a patient fails on a DTG-containing regimen with an increased risk for the emergence of resistance to INSTIs and the loss of future treatment options [7]. Given that TDF/TAF still exhibits full efficacy in the presence of the M184V/I mutation, this backbone component should be preferred in combination with DTG in low—and middle-income countries.

Our case has several limitations. Firstly, DTG drug levels were not measured at the time of virological failure and therefore suboptimal adherence as an important co-factor for the virological failure cannot be proven. However, self-reported adherence of missed doses of ART—as systematically collected in these patients within the Swiss HIV Cohort Study—is a good predictor for poor adherence and associated with an increased risk of both viral failure and death [14]. Hence, monitoring adherence helps to identify patients at risk for negative clinical outcomes and offers opportunities for intervention. Another limitation is that we didn’t sequence the integrase gene at baseline when the patient started his first cART, thus, we cannot rule out pre-existing INSTI-associated mutations. However, a recent large study within the SHCS clearly indicated that resistance-associated mutations to INSTIs were almost absent in Switzerland in drug naïve patients [15]. Finally, we were unable to perform next-generation sequencing of the virus from different time points because there was no plasma left for extraction. Therefore, we cannot exclude that the M184I mutation detected at different time points may potentially be linked to different viral quasispecies. However, while this discrimination is interesting from a viral evolution point of view, such information most likely would not have had any impact on the clinical management of the patient.

## 4. Conclusions

In conclusion, we describe a rare case of a virological failure on the DTG/3TC/ABC regimen with the emergence of resistance to all INSTIs in a patient with pre-existing M184V/I mutation. Other risk factors, which may have triggered the virological failure, included suboptimal adherence and a low nadir CD4+ cell count. This case illustrates that DTG/3TC/ABC should be prescribed with caution to patients with pre-existing M184V/I mutation, a low CD4 cell count, and/or poor adherence. Even more caution in this patient population is probably indicated when DTG/3TC containing dual regimens are planned. In addition, this case questions the current understanding of DTG as a drug with high resistance barrier similar to contemporary boosted PIs and highlights the need for close viral load monitoring in countries with a high prevalence of NRTI-mutations.

## Figures and Tables

**Figure 1 viruses-12-01330-f001:**
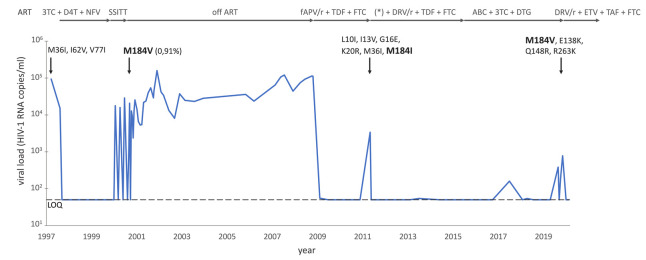
Virological evolution and emergence of resistance-associated mutations (arrows pointing downwards) of our patient since initiation of antiretroviral treatment. Abbreviations: ART: antiretroviral treatment, 3TC: lamivudine, D4T: stavudine, NFV: nelfinavir, SSITT: Swiss Spanish Intermittent Treatment Trial, FAPV/r: fosamprenavir/ritonavir, TDF: tenofovir disoproxil-fumarate, FTC: emtricitabine, (*) add-on raltegravir for one month, DRV/r: darunavir/ritonavir, ETV: etravirine, TAF: tenofovir alafenamide, LOQ: level of quantification. The persisting M184V/I mutation is highlighted in bold.

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
