# Peer review of "Emergence of Resistance to Integrase Strand Transfer Inhibitors during Dolutegravir Containing Triple-Therapy in a Treatment-Experienced Patient with Pre-Existing M184V/I Mutation"

_viruses, 2020, doi:10.3390/v12111330_

Round 1

Reviewer 1 Report

This is a very interesting and to my knowledge unique case report. I have just a few minor comments:

  1. I acknowledge that poor adherence can compromise any regimen; however, as the authors point out M184V/I decreases ABC susceptibility, but it enhances TDF/TAF susceptibility. Thus the impact of  M184V/I should more explicitly  state that  the triple regimen used in this patient was compromised while the regimen used in LMIC would have more potency.
  2. The figure does not make clear with mutations that appeared persisted and which did not. Importantly M184V was still present according to the text. Were any other NRTI mutations  present?
  3. There are a few typos and misspellings in the text.

Author Response

#Reviewer 1

This is a very interesting and to my knowledge unique case report. I have just a few minor comments:

I acknowledge that poor adherence can compromise any regimen; however, as the authors point out M184V/I decreases ABC susceptibility, but it enhances TDF/TAF susceptibility. Thus the impact of  M184V/I should more explicitly  state that  the triple regimen used in this patient was compromised while the regimen used in LMIC would have more potency.

Response: We thank the reviewer for addressing this important point. We added the information about the enhanced TDF/TAF susceptibility in the presence of M184V/I mutation in the manuscript.

It now says on page 2, lines 46ff:

The presence of M184V/I reduces the susceptibility to these drugs by more than 100-fold and additionally causes an impaired efficacy to abacavir (ABC); in contrast, it enhances the susceptibility to tenofovir disoproxil fumarate (TDF) and tenofovir alafenamid fumarate (TAF).

It now says on page 5, lines 164ff:

Given that TDF/TAF still exhibits full efficacy in the presence of the M184V/I mutation, this backbone component should be preferred in combination with DTG in low- and middle income countries.

The figure does not make clear with mutations that appeared persisted and which did not. Importantly M184V was still present according to the text. Were any other NRTI mutations present?

There are a few typos and misspellings in the text

Response: We added a sentence in the case description, highlighted the persisting M184V/I mutation in bold in the figure and updated the figure legend accordingly.

It now says on page 3, line 109:

In addition, the prior M184V mutation - but no other NRTI mutations - was detected at a frequency of 100%.

It now says in the figure legend:

The persisting M184V/I mutation is highlighted in bold.

Reviewer 2 Report

The case report by Braun et al, discusses the acquisition of drug resistance mutations in a patient on varying long-term ART regimens. The treatment was interrupted both in a planned manner, due to enrollment in the Swiss-Spanish Intermittent Treatment trial, as well as due to patient compliance failure. The patient had acquired the M184V mutation early in the treatment regimen. One of the main suggestions derived from the analysis is that the presence of pre-existing M184V/I mutations renders susceptibility to acquisition of INSTI resistance mutations, and that the resistance barrier to acquiring Dolutegravir (DTG) resistance mutations is perhaps lower than what is currently acknowledged.

The reviewer feels that this conclusion is not strongly supported by the data presented. Several studies have demonstrated that a high barrier to acquiring DTG resistance mutations in both 2 and 3 drug regimens in part due to DTG having a low inhibitory quotient and a wide therapeutic index. Low CD4 cell count coupled with poor adherence were likely compounding factors in the acquisition of INSTI resistance mutations. The authors acknowledge that since DTG levels were not measured, virological failure (coupled with the appearance of E138K, Q148R and R263K) could well be due to poor drug adherence. This is especially relevant after the patient was switched to a single tablet DTG/3TC/ABC regimen and reported poor adherence and illicit drug use at this time. Since the sanger sequencing at earlier time points do not provide information on whole genome linked co-purifying mutations, it is unclear to what extent the M184I observed in 2011 is related to the M184V observed in 2020. The integrase gene was not sequenced prior to commencement of cART. The individual impact of low CD4 count, preexisting M184I, and poor drug adherence on virological failure and INSTI drug resistance selection are not deconvolved, considering the small data set and availability of only bulk sequencing data. More detailed sequencing analysis would be necessary to make the connection between pre-existing M184V/I and a predisposition to gain INSTI resistance mutations. 

Author Response

Reviewer 2

The case report by Braun et al, discusses the acquisition of drug resistance mutations in a patient on varying long-term ART regimens. The treatment was interrupted both in a planned manner, due to enrolment in the Swiss-Spanish Intermittent Treatment trial, as well as due to patient compliance failure. The patient had acquired the M184V mutation early in the treatment regimen. One of the main suggestions derived from the analysis is that the presence of pre-existing M184V/I mutations renders susceptibility to acquisition of INSTI resistance mutations, and that the resistance barrier to acquiring Dolutegravir (DTG) resistance mutations is perhaps lower than what is currently acknowledged.

The reviewer feels that this conclusion is not strongly supported by the data presented. Several studies have demonstrated that a high barrier to acquiring DTG resistance mutations in both 2 and 3 drug regimens in part due to DTG having a low inhibitory quotient and a wide therapeutic index. Low CD4 cell count coupled with poor adherence were likely compounding factors in the acquisition of INSTI resistance mutations. The authors acknowledge that since DTG levels were not measured, virological failure (coupled with the appearance of E138K, Q148R and R263K) could well be due to poor drug adherence. This is especially relevant after the patient was switched to a single tablet DTG/3TC/ABC regimen and reported poor adherence and illicit drug use at this time. Since the sanger sequencing at earlier time points do not provide information on whole genome linked co-purifying mutations, it is unclear to what extent the M184I observed in 2011 is related to the M184V observed in 2020. The integrase gene was not sequenced prior to commencement of cART. The individual impact of low CD4 count, preexisting M184I, and poor drug adherence on virological failure and INSTI drug resistance selection are not deconvolved, considering the small data set and availability of only bulk sequencing data. More detailed sequencing analysis would be necessary to make the connection between pre-existing M184V/I and a predisposition to gain INSTI resistance mutations.

Response: The reviewer questions one of the main conclusions of our case description, namely that the resistance barrier to acquiring Dolutegravir (DTG) resistance mutations is maybe lower than what is currently acknowledged. We agree that so far only a few cases deriving from real life data report emergence of resistance to DTG among patients on DTG-containing triple therapy. We explicitly stated this paucity of data in the introduction section (page 1, line 35ff).  However, we feel that the data provided in our case description (detection of M184V/I mutation on several time-points in combination with poor adherence and low CD4 cell nadir) may explain the virological failure leading to resistance to DTG. This assumption is supported by the fact that longitudinal work from the Swiss HIV Cohort Study shows that resistance to INSTIs is almost absent in treatment-naïve HIV-infected patients. Hence, the presence of baseline resistance associated mutations to INSTIs is very unlikely in our patient (1, 2). In addition, we just published that pre-existing minor INSTI resistance mutations did not have an effect on treatment outcome in treatment naïve patients (3).  Unfortunately, the blood samples which would be needed to perform in depth sequencing analyses from our patient at different time-points are not available anymore. Because a final proof of our hypothesis based on the data available cannot be done, we tempered our statement about the factors leading to emergence of resistance throughout the manuscript and added the limitation as suggested by the reviewer.

It now says in the abstract:

Additional risk factors which may have triggered the virological failure included suboptimal adherence and low nadir CD4+ cell count.

It now says in the discussion (page 4, line 131ff)

Our patient exhibited several features that may have increased his risk for a virological failure

It now says in the discussion (page 5, line 151ff)

The learning points of this case are manifold: Firstly, this case illustrates that patients with a low CD4+ cell count nadir, pre-existing NRTI-associated mutations, and poor adherence are not candidates for a DTG-containing regimen with limited efficacy of the NRTI-backbone, at least not without close monitoring of the viral load.

It now says in the discussion (page 5, line 171ff)

Our case has several limitations.

It now says in the discussion (page 5, line 180ff)

Finally, we were unable to perform next generation sequencing of the virus from the different time points because there was no plasma left for extraction. Therefore, we cannot exclude that the M184I mutation detected at different time points may potentially be linked to different viral quasispecies. However, while this discrimination is interesting from a viral evolution point of view, such information most likely would not have had any impact on the clinical management of the patient. 

It now says in the conclusion (page 5, line 186ff)

Additional risk factors which may have triggered the virological failure included suboptimal adherence and a low nadir CD4+ cell count.

Literature:

  1. Scherrer AU, von Wyl V, Yang WL, Kouyos RD, Boni J, Yerly S, Klimkait T, Aubert V, Cavassini M, Battegay M, Furrer H, Calmy A, Vernazza P, Bernasconi E, Gunthard HF, Swiss HIVCS, Swiss HIVCS. Emergence of Acquired HIV-1 Drug Resistance Almost Stopped in Switzerland: A 15-Year Prospective Cohort Analysis. Clin Infect Dis. 2016;62(10):1310-7.
  2. Scherrer AU, Yang WL, Kouyos RD, Boni J, Yerly S, Klimkait T, Aubert V, Cavassini M, Battegay M, Hauser C, Calmy A, Schmid P, Bernasconi E, Gunthard HF, Swiss HIVCS. Successful Prevention of Transmission of Integrase Resistance in the Swiss HIV Cohort Study. J Infect Dis. 2016;214(3):399-402.
  3. Pyngottu A, Scherrer AU, Kouyos R, Huber M, Hirsch H, Perreau M, Yerly S, Calmy A, Cavassini M, Stockle M, Furrer H, Vernazza P, Bernasconi E, Gunthard HF, Swiss HIVCS. Predictors of virological failure and time to viral suppression of first line integrase inhibitor based antiretroviral treatment. Clin Infect Dis. 2020.

Round 2

Reviewer 2 Report

The revised manuscript acknowledging the caveats provides a more balanced interpretation of the results. The results would be of interest for researchers investigating emergence of drug resistance mutations in HIV.